# Assessing the impact of professional lactation support frequency, duration and delivery form on exclusive breastfeeding in Lebanese mothers

**Dayane Daou[1], Hani Tamim[2], Mona Nabulsi[3]***

**1** Department of Anesthesiology, American University of Beirut, Beirut, Lebanon, **2** Biostatistics Unit, Faculty of Medicine, Clinical Research Institute, American University of Beirut, Beirut, Lebanon, **3** Department of Pediatrics and Adolescent Medicine, American University of Beirut, Beirut, Lebanon

* mn04@aub.edu.lb

**Data Availability Statement:** All relevant data are within the supporting information file.

**Funding:** The authors received no specific funding for this work.

## Abstract

### Background

The optimal frequency, duration, and form of professional lactation support needed to continue exclusive breastfeeding (EBF) for six months have not yet been specifically identified. This study investigates the association between six-month EBF and the frequency, duration, and form (face-to-face vs. telephone contact) of professional lactation support in a cohort of Lebanese mothers, and explores barriers to EBF during the first six months postpartum.

### Methods

An observational study was nested in a breastfeeding support randomized controlled trial. Secondary analysis of data from 159 trial participants who received professional lactation support was conducted. (1) Six-month EBF with professional lactation support frequency, duration, and form was investigated using bivariate and multivariate regression analyses. (2) Barriers to breastfeeding were explored using content analysis of narrative data collected about breastfeeding mothers by the lactation experts.

### Results

Six-month EBF was achieved by 57/159 (35.8%) participants. Professional lactation support was received by more mothers continuing six months of EBF (100% vs. 83.3%, $p = 0.001$). In crude analysis, those mothers had a higher number of overall contacts with lactation experts (mean ± SD of 9.5 ± 2.9 vs. 7.0 ± 4.4, $p = 0.001$), and longer duration of face-to-face contacts (mean ± SD of 261.9 ± 209.1 vs. 201.0 ± 117.4 minutes, $p = 0.035$). In adjusted analysis, frequencies of overall and of telephone contacts with the lactation experts were positively associated with six-month EBF (OR = 1.15; *95% CI*: 1.04 to 1.27, $p = 0.007$; OR = 1.12; *95% CI*: 1.00 to 1.26, $p = 0.05$; respectively). Participants discontinuing EBF early were described as inexperienced, lacking breastfeeding knowledge, concerned about milk insufficiency, and showing negative attitudes towards night feeds.

**Competing interests:** The authors have declared
that no competing interests exist.

## Conclusion

Need-based telephone contact augmenting face-to-face professional lactation support may
positively influence six-month EBF. Early identification of mothers at risk for early discontin-
uation of EBF can help tailor interventions specific to their concerns.

## Introduction

Breastfeeding is the optimal infant nutrition contributing to reduced child under-five mortal-
ity and morbidity. Longer duration of breastfeeding is associated with lower risks for infec-
tions and chronic diseases, and with higher intelligence in children, as compared to shorter
duration [1]. Breastfeeding mothers have reduced risks for breast and ovarian cancers, and for
chronic diseases [1]. The World Health Organization (WHO) recommends exclusive breast-
feeding (EBF) for the first six months, with continuation of breastfeeding until at least two
years, supplemented with appropriate complementary foods. EBF is defined as feeding the
baby mother's milk only, with no other food or drink including water, but allowing oral rehy-
drating solutions, vitamins, minerals or other medicines when needed [2].

Lebanon, an upper middle income country has a low six-month EBF rate of 15% [3]. Barri-
ers to EBF in Lebanon include maternal, socio-cultural, and legislative factors. In a qualitative
study that followed mothers for one year after delivery, maternal perception of low milk sup-
ply, lack of family support, sleep deprivation, sore nipples, painful breastfeeding, and breast
engorgement were identified as important barriers to breastfeeding continuation [4]. More-
over, Bou Diab and Werle in a more recent qualitative study found that 'descriptive norms' of
the close community members and society at large may affect a mother's decision to breastfeed
or not as she would want to conform to these norms [5]. At the national legislative level, there
is poor dissemination, implementation, and enforcement of policies and laws that protect
breastfeeding [6], such as those pertaining to maternal employment, Baby Friendly Hospital
Initiative (BFHI) [7], and the implementation of the International Code of Marketing of Breast
milk Substitutes [6]. Moreover, hospitals and maternity clinics in Lebanon do not comply with
the ten steps of BFHI, and health professionals who care for breastfeeding mothers lack the
WHO, and United Nations International Children's Emergency Fund (UNICEF) recom-
mended training in the prevention and treatment of breastfeeding problems [8, 9]. Also, there
are few International Board Certified Lactation Consultants (IBCLCs) in Lebanon, and they
are not integrated within the national health system.

There is strong evidence on the effectiveness of breastfeeding promotion and support inter-
ventions in improving breastfeeding initiation and continuation rates [10–13]. For example,
antenatal breastfeeding education, peer support, and professional lactation support, delivered
as single interventions or in combinations, increase EBF and any breastfeeding rates [10–12,
14]. We recently reported similar findings from our multi-center, randomized controlled trial
(RCT), in which a multi-component intervention of antenatal education, peer, and profes-
sional support improved six-month EBF in healthy Lebanese mothers [15]. Participants
allocated to the multi-component intervention were twice as likely to EBF for 6 months, com-
pared to controls (OR = 2.02; *95% CI*: 1.20 to 3.39). Participants in the experimental group
who complied with all three intervention components were six times more likely than controls
to continue EBF for six months (OR = 6.63; *95% CI*: 3.03 to 14.51). Of the three intervention
components, participants were most compliant with professional lactation support (92.6%),
received either as a single component, or in combination with prenatal education and/or peer
support.

Despite the evidence on the effectiveness of professional lactation support in improving breastfeeding rates, there still remains a knowledge gap surrounding the optimal dose (frequency and duration), and form of support delivery needed to improve six-month EBF rates [13, 14]. In this study, we aimed to (1) investigate the association between professional lactation support (exposure) and six-month EBF (outcome), in terms of the support frequency, duration, or form of delivery (face-to-face versus telephone support), and to (2) explore the differences in breastfeeding barriers and facilitators between participants who continued six months of EBF, and those who stopped earlier. Face-to-face support is defined as support delivered during visits to the hospital or to the homes of the participants.

## Materials and methods

### Design

This was a retrospective case control observational study in which data that were previously collected in our RCT were analyzed. The trial enrolled healthy pregnant women who were randomly allocated to a control group that received standard prenatal and postpartum medical care, or to an experimental group that received a multi-component intervention consisting of prenatal breastfeeding education, peer support, and postpartum professional lactation support that was provided by IBCLCs. The primary outcome of the trial, exclusive breastfeeding (EBF), was assessed at the six month interview with the participants. More details on the trial can be found elsewhere [15]. The trial was approved by the Institutional Review Boards of the American University of Beirut and the Sahel General Hospital, and written informed consent was obtained from all participants prior to enrolment. In this observational study, cases were the trial participants who continued EBF for six months and controls were those who stopped EBF earlier than six months. This report conforms to the Strengthening the Reporting of Observational Studies in Epidemiology (STROBE) reporting guideline [16].

### Setting

Lebanon's health system is highly privatized, with practices relating to prenatal, postnatal care, and breastfeeding depending completely on individual care providers. Breastfeeding education is not part of prenatal care, and is provided postpartum by pediatricians and hospital nurses. Most hospitals separate mothers and infants after birth, and infant formula is provided to infants in nurseries. There are currently few IBCLCs in Lebanon. They work in private practices and not in hospitals. This health care system may contribute to the low breastfeeding rates in the country.

The RCT was conducted in two tertiary care hospitals in Beirut between December 2013 and January 2016, referred to herein as site 1 and site 2. Whereas site 1 was a large teaching private referral center located in the capital Beirut that serves middle and high income patients from different regions of the country, site 2 was a smaller tertiary hospital located in the southern suburb of Beirut, and caters mainly to the Muslim communities residing in its neighborhood characterized by middle to low socioeconomic status. Optional prenatal classes about labor and delivery are available at site 1 but not site 2, with minimal breastfeeding education. There are no official statistics on the breastfeeding rates in either site.

### Sample

Participants whose data were analyzed in this study belonged to the RCT's experimental arm (*N* = 174). They were healthy pregnant women who were in their first or second trimester, and who expressed interest in breastfeeding their infants. Women who were beyond the second

trimester, suffered from any chronic medical condition, had fetal abnormalities on routine mid-pregnancy obstetrical ultrasound, had twin gestation, were determined to use infant formula as the sole infant nutrition, were leaving Lebanon before six months, or delivered prior to 37 weeks of gestation were not enrolled in the RCT. Of the total sample size that was randomized to the experimental group (*N* = 174), 15 (8.6%) had missing data on infant nutrition at six months and were excluded from analysis, which represents an acceptable attrition rate.

## Description of the intervention

The RCT's participants who were allocated to the multi-component breastfeeding support intervention attended an antenatal breastfeeding education session that was delivered by an IBCLC soon after enrolment. Additionally, they received a booklet that summarized the benefits of breastfeeding, what to expect while breastfeeding, the different breastfeeding positions, and the hazards of human milk substitutes. Moreover, they were given a video that featured several misconceptions about breastfeeding (e.g., breastfeeding is painful and tiring to the mother, breastfeeding in public is shameful, mother's milk is not enough, avoidance of certain foods while breastfeeding, and others).

On the first day after delivery, participants were visited by peer supporters. These were volunteer women who delivered the peer support component of the trial's multi-component intervention. Peer supporters had to have a history of successive breastfeeding of at least one child for two or more months. Peer supporters attended two half-day training workshops soon after their enrolment in the trial. Peer support was delivered according to a pre-specified schedule (at least 10 telephone calls or hospital/home visits) until six months postpartum, or until the breastfeeding mother withdrew, or discontinued breastfeeding, whichever came first. Additional peer support was permitted if requested by the breastfeeding mother.

The third component of the multi-component intervention was professional lactation support, which was also started the first day after delivery. It was delivered as face-to-face IBCLC visits to the participants both in the hospital, and at home later. Home visits occurred on days three, seven, and fifteen after delivery, and then monthly until six months. Additional face-to-face visits or telephone support were allowed if needed. They could be initiated by the breastfeeding mother, or by the IBCLC whenever the need arose.

## Measurement

**Exposure.** Information on the four different exposures of professional lactation support was extracted from the IBCLC logbooks. The exposures of interest were the form (face-to-face vs. telephone calls) of lactation support, frequency of face-to-face and telephone contact, and duration of face-to-face and telephone contact.

**Outcome.** The primary outcome was EBF at six months as retrieved from the trial's SPSS dataset, and was validated against the IBCLC logbook documentation of infant nutrition. The secondary outcome was the difference in breastfeeding barriers and facilitators between participants who continued six months of EBF and those who stopped earlier, as retrieved from the IBCLC logbook narrative data.

**Confounders.** Information on potential confounders was retrieved from the trial's SPSS dataset. These included data on maternal age, site, gender of the obstetrician, age of the obstetrician (less or equal to 50 years vs. more than 50 years), gestational age at recruitment, maternal employment, religion, maternal education, monthly household income, number of children, number of breastfed children, previous longest breastfeeding duration, received antenatal breastfeeding education (yes/no), received peer support (yes/no), family support (yes/no), maternal breastfeeding knowledge score as assessed by the validated Arabic Infant

*Breastfeeding Knowledge Questionnaire* (BFK-A) at entry into the trial [17], and maternal breastfeeding attitude score as assessed by the validated Arabic version of the *Iowa Infant Feeding Attitude Scale* (IIFAS-A) [18].

## Data collection

Data on the RCT participants were collected between December 2013 and January 2016. Written informed consent was obtained from all participants. During each contact with the participant, IBCLCs recorded in special logbooks information about the form of contact (face-to-face vs. telephone call), the duration of contact in minutes, and the support details provided during that contact. This included provision of treatment for breast engorgement, sore nipple, infection, or assistance with positioning, latching, pumping, or others. Moreover, the IBCLCs documented in a narrative form the details of the breastfeeding problem, maternal concerns, and the advice given to the mother, and whether referral to a physician was done.

## Data analysis

**The association between professional lactation support and 6-month EBF.** Continuous data were summarized as means and standard deviations (*SD*), or as medians and interquartile ranges (*IQR*) when appropriate; whereas categorical data were summarized as frequencies and proportions. We conducted bivariate analysis to explore the association between the primary outcome (EBF at 6 months), and the various exposures (lactation support frequency, duration, and form of delivery) using Chi square test for categorical variables, and Student's t test for continuous variables. Similar bivariate analysis was performed to explore the association between the primary outcome and each of the potential confounders. Moreover, we conducted multivariate logistic regression analyses using the forward method to investigate the associations between the primary outcome (dependent variable), and the exposures (predictors), adjusting for confounders found to be significant in bivariate analysis. The SPSS version 24 was used for data entry, management, and analysis. Statistical significance was set at *p* value $< 0.05$. This study was approved by the Institutional Review Boards of both sites, and is registered in Current Controlled Trials ISRCTN17875591. www.isrctn.com

**Breastfeeding barriers and facilitators.** Details of maternal concerns or problems, advice or treatment provided by the IBCLC, and any other IBCLC observations were documented by the IBCLCs in narrative form in the logbook at each contact with the breastfeeding mother. These qualitative data were explored using content analysis aiming at identifying barriers to breastfeeding during the first six months. Two authors (DD and MN) independently read and analyzed the comments of the IBCLCs about their discussions with, and observations of the participants during home visits independently. Inductive manifest analysis [19] was done by first breaking down the data into smaller *meaning units*, and assigning *codes* to similar units. The codes were then compared against the original data, and homogeneous codes were grouped into *themes* as common concepts emerged from the data. The themes were subdivided by their time of occurrence: first postpartum week, second week to third month postpartum and fourth to sixth month postpartum. This temporal grouping was done to explore whether maternal breastfeeding concerns or challenges vary by time from delivery. The two authors compared the results of their analyses and resolved discrepancies by discussion until reaching consensus to improve on the credibility of the findings. To identify the differences in breastfeeding barriers and facilitators between participants who breastfed exclusively until six months, and those who stopped earlier, we compared themes generated from their IBCLC logbooks at each time interval.

## Results

### Characteristics of the sample

Of 159 mothers, 57 (35.8%) were exclusively breastfeeding at six months. Most participants (102 primi-gravid and 9 multi-gravid participants; 69.8%) had never breastfed previously. Tables 1 and 2 compare the baseline characteristics and breastfeeding support variables of participants who continued exclusive breastfeeding for six months, and those who stopped earlier. As compared to participants who stopped EBF early, mothers who continued for six months were younger, had better breastfeeding knowledge, more positive attitudes towards breastfeeding, breastfed more children, and had previously breastfed for a longer duration (for participants who had a previous breastfeeding experience). Moreover, more participants who continued EBF for six months belonged to site 2, were Muslims, had female obstetricians, had younger obstetricians, and received antenatal breastfeeding education, peer support, or professional lactation support (Table 2).

**Table 1. Characteristics of the cohort (N = 159).**

| Variables | 6-m EBF | <6-m EBF | p |
|---|---|---|---|
| | n = 57 M (SD) | n = 102 M (SD) | |
| Age (years) | 28.7 (4.3) | 30.3 (4.5) | 0.036 |
| Previous longest breastfeeding duration (months) | 10.2 (6.3) | 5.6 (5.0) | 0.003 |
| Breastfeeding Knowledge score on BFK-16 | 11.8 (2.5) | 10.8 (2.3) | 0.011 |
| IIFAS-A score | 72.6 (5.1) | 66.7 (8.5) | <0.001 |
| Number of peer support telephone calls | 8.9 (6.6) | 6.5 (4.9) | 0.027 |
| Total duration of peer support calls (minutes) | 67.9 (92.0) | 50 (93.0) | 0.976 |
| Gestational age at recruitment (weeks) | 17.8 (4.3) | 17.4 (4.3) | 0.589 |

EBF = Exclusive breastfeeding; BFK-16 = 16-item Arabic Breastfeeding Knowledge questionnaire [17]; Breast feeding knowledge scores ranged from 4 to 16 with higher scores indicating better knowledge; IIFAS-A = Iowa Infant Feeding Attitude Scale-Arabic version (IIFAS-A) [18]; higher scores indicate more positive attitudes.

**Table 2. Frequency distribution of participants characteristics (N = 159).**

| Variables | 6-m EBF | <6-m EBF | p |
|---|---|---|---|
| | n = 57 n (%) | n = 102 n (%) | |
| Center: Site 1 | 46 (80.7) | 95 (93.1) | 0.018 |
| Male Gender of Obstetrician | 23 (40.4) | 66 (64.7) | 0.003 |
| Age of Obstetrician >50 years | 27 (47.4) | 65 (63.7) | 0.045 |
| Muslim Religion | 53 (98.2) | 77 (80.2) | 0.005 |
| University Education | 47 (82.5) | 84 (82.4) | 0.987 |
| Monthly household income >1000 USD | 41 (77.4) | 76 (81.7) | 0.387 |
| Number of children ≥1 | 29 (50.9) | 39 (38.3) | 0.079 |
| Number of breastfed children ≥1 | 28 (49.1) | 32 (31.4) | 0.021 |
| Has Family Support | 38 (95.0) | 64 (100) | 0.146 |
| Received Antenatal Education | 28 (49.1) | 25 (24.5) | 0.002 |
| Received Peer Support | 49 (86.0) | 69 (67.6) | 0.011 |
| Received Professional Lactation Support | 57 (100) | 85 (83.3) | 0.001 |

Missing values for Has Family Support = 55; EBF = Exclusive breastfeeding; Professional Lactation Support is defined as provision of lactation support by International Board Certified Lactation Consultants; Median monthly household income = 2546 USD.

## The association between professional lactation support and 6-month EBF

In bivariate analysis, participants who continued EBF for six months had a higher number of face-to-face contacts and telephone contacts with the IBCLCs. Also, the duration of face-to-face contacts with the IBCLCs was longer (Table 3).

In the multivariate adjusted analyses, the overall number of IBCLC contacts (face-to-face plus telephone) was associated with higher odds for six-month EBF (OR = 1.15, *95% CI*: 1.04 to 1.270), whereas face-to-face contacts alone were not. This suggests that home visits along with telephone contacts with the IBCLCs may be more effective than home visits alone. The number of IBCLC telephone contacts was associated with slightly higher odds for six-month EBF; OR = 1.12, *95% CI*: 1.00 to 1.26, respectively). Neither the number, nor the duration of face-to-face IBCLC contact was associated with six-month EBF. Other positive predictors of six-month EBF were receipt of antenatal breastfeeding education, and having a female obstetrician (Table 4).

## Breastfeeding barriers and facilitators

Content analysis of maternal concerns and breastfeeding problems as documented in the statements of the IBCLCs identified the following themes: Maternal and infant nutrition concerns, maternal and infant sleep problems, breastfeeding technical problems, maternal and infant sickness, and maternal or community attitudes towards breastfeeding. These concerns were common among participants who completed EBF for six months, and those who stopped earlier. The two groups however differed in the nature of the concerns or problems, their intensity, duration, or the way to resolve them.

**Maternal and infant nutrition.** During the first week postpartum, according to the IBCLCs, participants practicing six months of EBF were interested in "the best diet to increase maternal milk production". This interest continued through the first three months. By six months, mothers' main concerns shifted to how best to introduce complementary foods in addition to breastfeeding.

The IBCLCs reported that the main concerns of the participants discontinuing EBF early were that "mother's milk was not enough", "the baby was not satisfied", and "the baby does not have enough wet diapers". During the first three months, most participants were giving their infants infant formula, especially at night, in addition to breastfeeding. They were also adding rice to the infant formula. Mothers described their infants as "refusing the breast and enjoying the bottle", or "refusing frozen expressed mother's own milk". Reasons given by mothers for discontinuing EBF were "maternal milk is not enough", "baby has reflux",

**Table 3. Bivariate analysis of the association between six-month EBF and different exposures (*N* = 159).**

| Exposure | 6-m EBF | <6-m EBF | *p* |
|---|---|---|---|
| | (*n* = 57) | (*n* = 102) | |
| | *M (SD)* | *M (SD)* | |
| Number of overall IBCLC contacts | 9.5 (2.9) | 7.0 (4.4) | <0.001 |
| Number of face-to-face IBCLC contacts | 4.5 (2.9) | 3.1 (2.6) | 0.003 |
| Number of IBCLC telephone contacts | 5.0 (3.0) | 3.9 (3.2) | 0.031 |
| Duration of overall IBCLC contacts (min) | 281.1 (209.8) | 226.0 (137.3) | 0.065 |
| Duration of face-to-face IBCLC contacts (min) | 261.9 (209.1) | 201.0 (117.4) | 0.035 |
| Duration of IBCLC telephone contacts (min) | 68.2 (43.9) | 68.6 (48.3) | 0.976 |

IBCLC = International Board Certified Lactation Consultant; min = Minutes; EBF = Exclusive breastfeeding.

**Table 4. Predictors of six-month exclusive breastfeeding (N = 159).**

| Predictors | OR (95% CI) | p |
|---|---|---|
| Model 1 | | |
| *Number of overall IBCLC contacts | 1.15 (1.04 to 1.27) | 0.007 |
| Site | 4.31 (1.37 to 13.56) | 0.012 |
| Obstetrician's gender | 2.87 (1.38 to 5.99) | 0.005 |
| Antenatal education | 2.98 (1.38 to 6.43 | 0.005 |
| Model 2 | | |
| *Number of face-to-face IBCLC contacts | 1.133 (0.99 to 1.30) | 0.071 |
| Site | 4.55 (1.50 to 13.77) | 0.007 |
| Obstetrician's gender | 2.83 (1.37 to 5.83) | 0.005 |
| Antenatal education | 2.85 (1.31 to 6.22) | 0.008 |
| Model 3 | | |
| *Number of IBCLC telephone contacts | 1.12 (1.00 to 1.26) | 0.05 |
| Site | 4.58 (1.49 to 14.12) | 0.008 |
| Obstetrician's gender | 2.96 (1.43 to 6.13) | 0.003 |
| Antenatal education | 3.73 (1.74 to 7.99) | 0.001 |
| Model 4 | | |
| *Overall duration of face-to-face IBCLC contact | 1.00 (0.99 to 1.00) | 0.311 |
| Site | 3.19 (0.97 to 10.54) | 0.057 |
| Obstetrician's gender | 3.12 (1.45 to 6.71) | 0.004 |
| Antenatal education | 2.75 (1.20 to 6.31) | 0.017 |

*Forced variable; Variables in the 4 models = Age, obstetrician's gender, obstetrician's age, site, breastfeeding knowledge score, antenatal education, peer support; IBCLC = International Board Certified Lactation Consultant.

mastitis, travel, maternal employment, or a pediatrician's advice to "start infant formula to increase the baby's weight". Few mothers seemed to be concerned about their figure and their own diet as reported by the IBCLCs.

**Maternal and infant sleep.** In the first week, participants with six months of EBF were concerned about "when to wake up the baby at night to breastfeed", and the "frequency of night feeds". Participants who stopped EBF earlier were reported by the IBCLCs to express negative feelings towards night feeding that persisted through the first three months. Both groups reported better infant sleeping patterns after three months.

**Breastfeeding technical problems.** The IBCLCs reported that in the first week, participants practicing EBF for six months were concerned with storage of mother's own milk; pump sterilization, manual expression of mother's milk, and cup feeding. This was to assure having enough of mother's own milk for the infant when she is unavailable (e.g., when mother returns to the work place, or when feeding the infant if present in public places since breastfeeding in public is culturally unacceptable). Later, the main concern was how to maintain milk production after returning to work. In the last three months, mothers reported excellent pumping results. Participants stopping EBF earlier were described by the IBCLCs as reporting more difficulties with breastfeeding positions and latching techniques in the first week; and breast engorgement, sore nipples, and mastitis during the first three months.

**Maternal and infant sickness.** Participants discontinuing EBF early reported more infant health issues in the first week like admissions to the Neonatal Intensive Care Unit and jaundice, and more maternal health issues. Between the second week and three months, more infants were reported to have reflux, heart problems, or viral infections, some of which necessitated hospital admission. Both groups had no health issues after three months.

**Attitudes towards breastfeeding.** The IBCLCs described participants who continued EBF for six months as "showing a lot of excitement" about breastfeeding in the first week. They were "self-confident", "more experienced", "more determined to exclusively breastfeed", and "very willing to continue as much as they could". During the first three months, they were reported as being more "comfortable and relaxed", "very excited", and "satisfied with the pumping results". By six months, they were "very satisfied with the whole breastfeeding experience", describing it as a "great achievement".

In contrast, participants who discontinued EBF early were described by the IBCLCs as "anxious", "emotional", "distressed about breastfeeding", with "low self-confidence" during the first week. During the first three months, they were reported as "uncooperative", showing "frustration and stress", "not confident" and "anxious" about their infants' sleep and nutrition patterns. Some refused to be visited or contacted by the IBCLC. The consultants described them as "not convinced about breastfeeding" since the beginning, an attitude that persisted through the first three months. Regardless of the breastfeeding outcome, most participants in this group were "thankful" for the follow up and assistance of the IBCLCs. Interestingly, mothers who stopped EBF earlier than six months had lower IIFAS-A scores than those who continued breastfeeding indicating more negative attitude towards breastfeeding (Table 1).

It is worth mentioning that both groups of participants repeatedly worried about societal influences (e.g., "society won't accept women to breastfeed in public").

## Discussion

In this observational study, we explored the association between the frequency, duration, and form of professional lactation support and six months exclusive breastfeeding. McFadden, et al. [13] reported that more than four postnatal contacts with professional lactation experts seemed to be most influential in reducing the odds of cessation of EBF before six months (OR = 0.89; *95% CI*: 0.81 to 0.98). It is interesting to note that both of our groups had more than four overall contacts with the IBCLC, which implies that the additional telephone contact may have contributed to the success of EBF until six months. Lavender, et al. [20] reported that telephone support reduced the risk for cessation of EBF between three and six months by 50%, as compared to any other support, or to no telephone support (RR = 1.5; *95% CI*: 1.19 to 1.93). The difference in IBCLC telephone support between our two groups could be because participants who were determined to continue EBF were more motivated to call the IBCLC when needed (as a need-based consultation), on top of utilizing more face-to-face contacts. This is further supported by the qualitative data reported by the IBCLCs in which they described participants who stopped EBF early as uncooperative and refusing IBCLC calls or visits. This may be because these participants were not convinced of breastfeeding, had more negative attitudes towards breastfeeding as evidenced by their lower IIFAS-A scores, or lacked self-confidence, and hence were distressed and frustrated with breastfeeding.

It is interesting to note that our participants who achieved six months of EBF had, in addition to more overall contacts with IBCLCs, better baseline breastfeeding knowledge than those who stopped earlier. This finding is in agreement with previous literature in which the combination of breastfeeding education and professional lactation support was reported to improve six-month EBF more than breastfeeding education alone (OR = 2.50; *95% CI*: 1.00 to 6.25) [11, 12]. We hypothesize that participants with better baseline breastfeeding knowledge were more enthusiastic about breastfeeding, which may have motivated them to seek further knowledge by attending the antenatal breastfeeding educational session designed for the trial participants allocated to the experimental group. Hence, receipt of antenatal breastfeeding education in addition to having better baseline breastfeeding knowledge may have contributed to their

success in achieving six months of EBF. This is in line with the IBCLC reports about participants who continued six months of EBF showing more self-confidence, and being excited, more determined, and very willing to continue EBF as much as they could.

The association of the obstetrician's gender with six-month EBF has not been previously reported. However, a previous study from Lebanon reported a positive association between female gender of the pediatrician and breastfeeding for four months (OR = 1.49; *95% CI*: 1.03 to 2.15) [21]. Moreover, a study reported the odds of EBF at three months to be higher among women whose birth was attended by a midwife or female nurse, as compared to birth by an obstetrician (OR = 1.87; *95% CI*: 1.34 to 2.61) [22]. This may be because female obstetricians, who may have experienced motherhood and/or breastfeeding, would be more comfortable to address maternal concerns about infant nutrition, as compared to male obstetricians. Also breastfeeding mothers may feel more comfortable to discuss breastfeeding with female obstetricians. However, this needs to be better explored in future studies, given that the current recommendation for obstetricians is to advocate for, and support breastfeeding [23].

The finding that participants who were recruited from site 2 were more successful in achieving six months of EBF than those recruited from site 1 can be explained by the characteristics of the population at each site. When compared to site 1, site 2 serves mostly middle to low income Muslim communities. The Holy Quran asks women to breastfeed their infants for two years. Moreover, breastfeeding is more economic for families who have middle, or low income. Our finding is similar to a previous study from Lebanon that reported Muslim religion, and lower maternal and paternal education (as surrogates of lower SES), to positively predict breastfeeding for four months [21].

The association of six-month EBF with younger maternal age is in contrast to reports from Western countries where younger maternal age is associated with lower EBF rates [24]. We believe that our finding may be due to a generational change in attitude towards breastfeeding in Lebanon, in which younger women are more aware of breastfeeding and its benefits, and thus are more committed to EBF than older mothers.

Analysis of maternal breastfeeding concerns or problems, our study's second aim, identified similar concerns in the two groups of participants: those who continued six months of EBF, and those who stopped earlier. However, the two groups differed in the type of concern, its severity, and the way to cope with it. For example, whereas nutritional maternal concerns of participants continuing six months of EBF were about the best diet to increase or maintain their milk, those who stopped EBF earlier were concerned about diets that will help them lose weight and regain their shape. Similarly, participants who stopped EBF early continued to express concerns about insufficiency of maternal milk, and were frustrated with the frequent night feeds that they interpreted as lack of infant's satiety. Instead of breastfeeding on demand at night, they supplemented their infants with infant formula. This may explain why more participants in this group suffered from breast engorgement. The temporal grouping of themes generated from the qualitative data revealed that the first three months postpartum was a critical period for breastfeeding participants as they faced sleep deprivation, problems with positioning and latching, and struggled with nutritional concerns, sore nipples, breast engorgement and other medical problems. Participants who lacked self-confidence, good breastfeeding knowledge, and good lactation support easily succumbed to those challenges and discontinued breastfeeding early.

Comparison of the two groups of participants in this study helps us identify factors that describe those who are at risk of stopping exclusive breastfeeding before six months. Breastfeeding women who are inexperienced, have poor breastfeeding knowledge, express concerns about breastfeeding affecting their weight or shape, are anxious about their milk being insufficient for the infant, or have negative feelings towards night feeds are at high risk of

discontinuing EBF before six months. Other red flags include maternal or infant health problems during the first three months postpartum, and specifically during the first week. We and others have reported on those factors as important barriers to breastfeeding continuation [4, 5, 24, 25]. Early identification of breastfeeding women facing difficulties with breastfeeding will alert the health care professional, be it a physician, a midwife, a nurse or an IBCLC to the different challenges they are struggling with, especially during the first three months postpartum.

## Limitations

Since the study was observational, the association between the professional lactation support frequency, type, or duration with six-month EBF cannot be interpreted as causal. Causal associations require experimental designs like randomized controlled trials. It is however challenging to conduct randomized trials from an ethical perspective, given the value of professional lactation support for breastfeeding mothers. Another limitation is that the participants had intended initially to breastfeed, which was one of the inclusion criteria of the breastfeeding trial in which this study was nested. Hence this creates an unavoidable selection bias that limits generalizability of our findings to women who decline breastfeeding in the first place. Moreover, since our participants come from an urban setting, the findings cannot also be generalized to women in rural areas who might not be accepting of home visits or telephone calls from IBCLCs. Another limitation is the fact that the qualitative data is generated from statements made by the IBCLCs about their discussions with the mothers or observations made during their home visits. Hence the potential for conscious or unconscious subjective bias cannot be ruled out.

## Conclusion

Breastfeeding women may face several challenges during the first three months postpartum that may interfere with continuation of EBF until six months. The availability of close contact with professional lactation experts as face-to-face, complemented with unrestricted need-based telephone support may positively influence EBF for six months. Participants who continue to express concerns about milk insufficiency, weight concerns, night feeds, seem to be anxious, or lack self-confidence may be at risk of discontinuing EBF earlier than six months. Timely identification of these mothers by health professionals and lactation experts may help tailor interventions that specifically address their concerns, hence reducing the risk of early EBF cessation. Further research is needed to explore what interventions can effectively target those challenging concerns. Moreover, the superiority of face-to-face and need-based telephone lactation support over face-to-face support in our observational study may be skewed by selection bias; hence randomized clinical trials are needed to confirm this finding.

## Supporting information

**S1 Dataset. Anonymized data set.**
(SAV)

## Acknowledgments

This study constituted the dissertation subject of Dr. Dayane Daou as a requirement for her Masters in the Scholars in HeAlth Research Program (SHARP), at the American University of Beirut. We are thankful to Dr. Ghada El-Hage Fuleihan, and to Dr. Tamar Kasholian-Kabalian for their advice and critical comments on the manuscript. We are indebted to all our participants without whom this study could not have been possible.

## Author Contributions

**Conceptualization:** Dayane Daou, Mona Nabulsi.

**Data curation:** Mona Nabulsi.

**Formal analysis:** Dayane Daou, Hani Tamim, Mona Nabulsi.

**Investigation:** Dayane Daou, Mona Nabulsi.

**Methodology:** Dayane Daou, Hani Tamim, Mona Nabulsi.

**Project administration:** Mona Nabulsi.

**Resources:** Mona Nabulsi.

**Supervision:** Mona Nabulsi.

**Validation:** Hani Tamim, Mona Nabulsi.

**Visualization:** Mona Nabulsi.

**Writing – original draft:** Dayane Daou, Mona Nabulsi.

**Writing – review & editing:** Dayane Daou, Hani Tamim, Mona Nabulsi.

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
