## [Decision Letter · Decision Letter 0]

13 Jul 2020

PONE-D-20-14153

Assessing the impact of professional lactation support frequency, duration and delivery form on exclusive breastfeeding in Lebanese mothers

PLOS ONE

Dear Dr. Nabulsi,

Thank you for submitting your manuscript to PLOS ONE. After careful consideration, we feel that it has merit but does not fully meet PLOS ONE’s publication criteria as it currently stands. Therefore, we invite you to submit a revised version of the manuscript that addresses the points raised during the review process.

In addition to addressing the issues raised by the two reviewers please address the following:

Page 9 Line 175 Please identify in the description of the outcome variable whether the WHO definition was used to define EBF. If not please clarify how EBF was defined. 

World Health Organization (2008) *Indicators for assessing infant and young child feeding practices*. Geneva: World Health Organization. https://www.unicef.org/nutrition/files/IYCF_updated_indicators_2008_part_1_definitions.pdf

Page 11 The content analysis is very loosely described and does not appear to be informed by a theoretical model. Bengtsson M (2016) How to plan and perform a qualitative study using content analysis. *NursingPlus Open*
**2**, 8-14.

We look forward to receiving your revised manuscript.

Kind regards,

Jane Anne Scott, PhD, MPH Grad Dip Dietetics, BSc

Academic Editor

PLOS ONE

Journal Requirements:

Reviewers' comments:

Reviewer's Responses to Questions

**Comments to the Author**

1. Is the manuscript technically sound, and do the data support the conclusions?

Reviewer #1: Partly

Reviewer #2: Partly

2. Has the statistical analysis been performed appropriately and rigorously? 

Reviewer #1: Yes

Reviewer #2: I Don't Know

3. Have the authors made all data underlying the findings in their manuscript fully available?

Reviewer #1: No

Reviewer #2: Yes

4. Is the manuscript presented in an intelligible fashion and written in standard English?

Reviewer #1: Yes

Reviewer #2: Yes

5. Review Comments to the Author

Reviewer #1: Assessing the impact of professional lactation support frequency, duration and delivery form on exclusive breastfeeding in Lebanese mothers

PONE-D-20-14153

Introduction

[Page 4, Lines 62-36] BF associated lowered risks in mother or child, clarify? Please add references.

What do you mean by “intelligence quotient”?

Please add references to statements in lines 69-72, providing some studies with these findings.

[Page 6 Line 124-125] Please remove statements with personal opinions.

Methods:

I seem to miss how attitudes were measured? Please state how attitudes were measured and if this extraction method described in the methodology is reliable and unbiased?

Results

[Page 16, Lines 269-274] You need to provide more details on how attitudes differed between those participants who completer EBF for six months and those who stopped earlier, as it is your main outcome according to objective 2.

I don’t know how maternal and infant nutrition, sleep and technical problems, sickness, fit into your objectives? Please clarify in your introduction and methods.

Reviewer #2: Review

Thank you for the opportunity to review this interesting paper. Breastfeeding is an important public health topic and any research that improves our understanding of how to improve support for women is to be welcomed. In particular, there is a dearth of research from the WHO Eastern Mediterranean Region.

The manuscript is well written and provides secondary analysis of data from a randomised controlled trial.

Some revisions are needed before it is acceptable for publication.

The analysis is described correctly as an observational study but it is not a cohort study. As I understand it, the two groups were selected on the basis of an outcome i.e. exclusive breastfeeding for 6 months or not. The analysis addressed exposure to different levels of breastfeeding support from IBCLCs. Thus it is a retrospective case control study with women who breastfed exclusively for 6 months as the cases and those who did not are the controls. The methods section needs to be revised to make the groups clearer, particularly in the description of the sample.

The content analysis was based on logs kept by the IBCLCs. The way the findings are reported suggests that there has been some comparison of the two groups but it is not clear how this was undertaken or measured. For example, in Line 280-1 you state that women who discontinued EBF before 6 months were more concerned about their figure and their own diet - but how did you measure this? A clearer description of the methods and a more cautious presentation of the findings is needed. In addition, the findings and the discussion give the impression that this data came from women themselves rather than from the IBCLCs. Again this needs to be clearer. The discussion should use statements such as According to IBCLCs..... rather than statements about what women do and don't do. This also should be highlighted as a limitation of the study. Is it possible that there may be (conscious or unconscious) bias in the IBCLCs accounts? Especially as you suggest that the IBCLCs described some women as 'uncooperative'. The interpretation of the findings therefore needs to be more cautious

Other minor revisions needed:

Abstract

The findings not very informative as they don’t state what frequency, duration etc are associated with the outcome.

Introduction

Line 77: Please use the correct name for the International Code i.e. International Code of Marketing of Breastmilk Substitutes and it should be capitalized.

Lines 90-96 are confusing - what is the difference between participants in the multi-component arm and participant in all 3 intervention components? And what does ‘most engaged with’ mean? Please clarify in the paper.

For clarity please refer to breastfeeding mothers (not nursing mothers) throughout the paper.

Materials and methods

Line 122 – please change artificial milk to either BMS or infant formula (and throughout the paper)

Line 123 – this is an ambiguous sentence – due you mean there are few IBCLCs who work in private practices and that most work in hospitals? Please clarify.

Lines 150-1 I am not sure these can be described as cultural misconceptions – many women all over the world experience breastfeeding as painful and tiring; and as shameful because of the reaction of others. Please reword to reflect this.

Lines 220-1 – the categories not mutually exclusive - in which category does 3 months belong?

Results

In Table 1 It is unclear what the gestational age row refers to? As it is 17 weeks it can't be at birth - is it at recruitment? Please clarify.

You use terms face-to-face and home visits - are these the same? Please explain your use of the terms.

Discussion

Lines 344-6 – this seems to be speculation – is there other evidence you can provide to support your claims?

Line 365-7; It could also be that women feel more comfortable to discuss breastfeeding with female obstetricians.

6. PLOS authors have the option to publish the peer review history of their article (what does this mean?). If published, this will include your full peer review and any attached files.

Reviewer #1: No

Reviewer #2: **Yes: **Alison McFadden

---

## [Author Response · Author response to Decision Letter 0]

12 Aug 2020

Response to Editorial comments 

1. Add to the beginning of the Methods the names of the IRBs that approved the study, and whether informed consent was obtained.

Answer: We added the following sentence to the first paragraph of the Methods: “The trial was approved by the Institutional Review Boards of the American University of Beirut and the Sahel General Hospital, and written informed consent was obtained from all participants prior to enrolment”. 

2. Page 9 Line 175: Please identify in the description of the outcome variable whether the WHO definition was used to define EBF. If not please clarify how EBF was defined. 

Answer: We used the WHO definition of EBF. We added this information to the revised manuscript: “The World Health Organization (WHO) recommends exclusive breastfeeding (EBF) for the first six months, with continuation of breastfeeding until at least two years, supplemented with appropriate complementary foods. EBF is defined as feeding the baby mother’s milk only, with no other food or drink including water, but allowing oral rehydrating solutions, vitamins, minerals or other medicines when needed [2]”. 

2. Page 11: The content analysis is very loosely described and does not appear to be informed by a theoretical model. (See Bengtsson M, 2016).

Answer: Thank you very much for providing this valuable reference which helped us revise the reporting of the content analysis as follows: “These qualitative data were explored using content analysis aiming at identifying barriers to breastfeeding during the first six months. Two authors (DD and MN) independently read and analyzed the comments of the IBCLCs about their discussions with, and observations of the participants during home visits. Inductive manifest analysis [19] was done by first breaking down the data into smaller meaning units, and assigning codes to similar units. The codes were then compared against the original data, and homogeneous codes were grouped into themes as common concepts emerged from the data. The themes were subdivided by their time of occurrence: first postpartum week, second week to third month postpartum, and fourth to sixth month postpartum. This temporal grouping was done to explore whether maternal breastfeeding concerns or challenges vary by time from delivery. The two authors compared the results of their analyses and resolved discrepancies by discussion until reaching consensus to improve on the credibility of the findings”.

3. Please ensure that your manuscript meets PLOS ONE’s style requirements, including those for the naming. 

Answer: We reviewed and revised the manuscript to meet PLOS ONE’s style. We hope that you find no violations of the requirements. Should there be any that escaped our scrutiny, we would greatly appreciate if you can alert us of the specific deficiency in future communications so we address it immediately and correctly. 

Answer: We submitted our anonymized dataset to PLOS ONE with the revised manuscript as Supporting Information File (S1 Dataset). Kindly update our Data Availability statement on our behalf to reflect this information.

Response to Reviewer #1

We greatly appreciate your positive comments and suggestions. Kindly find below our response. 

1. Introduction, Page 4 Lines 62-63: BF associated lower risks in mother or child, clarify? Please add references.

Answer: We revised the sentence as follows: “Longer duration of breastfeeding is associated with lower risks for infections and chronic diseases and with higher intelligence in children, as compared to shorter duration [1]”. Kindly note that the provided reference summarizes the findings of 28 systematic reviews on the short- term and long-term effects of breastfeeding on children in low, middle and high income countries. 

2. Introduction, Page 4 Line 64: What do you mean by “intelligence quotient”? 

Answer: We removed the word “quotient” from this sentence. 

3. Introduction, Page 4 Lines 69-72: Please add references to the statement providing some studies with findings. 

Answer: We revised this sentence as follows: “Barriers to EBF in Lebanon include maternal, socio-cultural, and legislative factors. In a qualitative study that followed mothers for one year after delivery, maternal perception of low milk supply, lack of family support, sleep deprivation, sore nipples, painful breastfeeding, and breast engorgement were identified as important barriers to breastfeeding continuation [4]. Moreover, Bou Diab and Werle in a more recent qualitative study found that ‘descriptive norms’ of the close community members and society at large may affect a mother’s decision to breastfeed or not as she would want to conform to these norms. [5]. At the national legislative level, there is poor dissemination, implementation, and enforcement of policies and laws that protect breastfeeding [6]”.

4. Introduction, Page 6 Lines 124-125: Please remove statements with personal opinions.

Answer: We removed the word “fragmented” from this sentence. 

5. Methods: I seem to miss how attitudes were measured? Please state how attitudes were measured and if this extraction method described in the methodology is reliable and unbiased.

Answer: The attitudes of the breastfeeding mothers were measured by the validated Arabic Iowa Infant Feeding Attitude Scale (Charafeddine et al., J Hum Lact. 2016;32:309-314). The peer supporters were interviewed by our research team upon recruitment and were queried about their previous history of breastfeeding to decide whether they would meet our inclusion criteria. We agree with the kind reviewer, this way of assessing attitudes of peer supporters is not as objective as using a validated instrument to measure attitudes and hence is prone to bias. We thus removed this sentence from the revised manuscript and revised it as follows: “Peer supporters had to have a history of successive breastfeeding of at least one child for two or more months”. 

6. Results, Page 16 Lines 269-274: You need to provide more details on how attitudes differed between those participants who completed EBF for 6 months and those who stopped earlier, as it is your main outcome according to objective 2. 

Answer: We added to the Methods, Confounders, the following sentence: “maternal breastfeeding attitude score as assessed by the validated Arabic version of the Iowa Infant Feeding Attitude Scale (IIFAS-A) [18]”. In the Results, Characteristics of the sample: “mothers who continued for six months…had more positive attitudes towards breastfeeding”; to Table 1 we added the attitude scores of the 2 groups and their p value (>0.001); and in Results, section Attitudes towards breastfeeding, we added: “Interestingly, mothers who stopped EBF earlier than six months had lower IIFAS-A scores than those who continued breastfeeding indicating more negative attitude towards breastfeeding (Table 1)”. In the Discussion, end of first paragraph we added: “This is further supported by the qualitative data reported by the IBCLCs in which they described participants who stopped EBF early as uncooperative and refusing IBCLC calls or visits. This may be because these participants were not convinced of breastfeeding, had more negative attitudes towards breastfeeding as evidenced by their lower IIFAS-A scores, or lacked self-confidence, and hence were distressed and frustrated with breastfeeding”. 

7. Results, Page 16 Lines 269-274: I don’t know how maternal and infant nutrition, sleep and technical problems, sickness, fit into your objectives. Please clarify in your Introduction and Methods.

Answer: The second aim of this study as stated in the last paragraph of the Introduction is to explore the differences in breastfeeding barriers and facilitators between participants who continued six months of EBF, and those who stopped earlier. We also cited in the Introduction previous literature from Lebanon that identified some barriers to breastfeeding in this country which included sleep deprivation, insufficient milk, breast engorgement, sore nipple, etc... The themes that were generated from the qualitative data analysis actually support the previous literature as we identified similar barriers in the group that discontinued breastfeeding early, despite the professional lactation support they received from the IBCLCs. 

Response to Dr. Alison McFadden (Reviewer #2)

Thank you for your critical review of our manuscript and the valuable comments. Please find below our point by point response. 

1. The analysis is described as an observational study but it is not a cohort study. As I understand it, the 2 groups were selected on the basis of an outcome, i.e exclusive breastfeeding for 6 months or not. The analysis addressed exposure to different levels of breastfeeding support from IBCLCs. Thus it is a retrospective case control study with women who breastfed exclusively for 6 months as the cases and those who did not are the controls. The Methods section needs to be revised to make the groups clearer, particularly in the description of the sample.

Answer: We agree with the kind reviewer. We revised the Methods, Design to describe the study as a case control study. We also added the following sentence: “In this observational study, cases were the trial participants who continued EBF for six months and controls were those who stopped EBF earlier than six months”. 

2. The content analysis was based on logs kept by the IBCLCs. The way the findings are reported suggests that there has been some comparison of the 2 groups but it is not clear how this was undertaken or measured. For example, in Lines 280-281 you state that women who discontinued EBF before 6 months were more concerned about their figure and their own diet – but how did you measure this? A clearer description of the methods and a more cautious presentation of the findings are needed. 

Answer: We revised the Methods, qualitative data analysis as follows: “These qualitative data were explored using content analysis aiming at identifying barriers to breastfeeding during the first six months. Two authors (DD and MN) independently read and analyzed the comments of the IBCLCs about their discussions with, and observations of the participants during home visits. Inductive manifest analysis [19] was done by first breaking down the data into smaller meaning units, and assigning codes to similar units. The codes were then compared against the original data, and homogeneous codes were grouped into themes as common concepts emerged from the data. The themes were subdivided by their time of occurrence: first postpartum week, second week to third month postpartum, and fourth to sixth month postpartum. This temporal grouping was done to explore whether maternal breastfeeding concerns or challenges vary by time from delivery. The two authors compared the results of their analyses and resolved discrepancies by discussion until reaching consensus to improve on the credibility of the findings”.

In the findings, since the comment about maternal figure and diet was reported by the IBCLCs based on maternal questions about the best diet to help them lose weight, we revised this sentence as follows: “Few mothers seemed to be concerned about their figure and their own diet as reported by the IBCLCs”.

3. The findings and the discussion give the impression that this data came from women themselves rather than from the IBCLCs. Again this needs to be clearer. The Discussion should use statements such as ‘According to IBCLC..’ rather than statements about what women do and don’t do. This also should be highlighted as a limitation of the study. 

Answer: We revised the findings and Discussion to reflect the fact that the findings were generated from the notes of the IBCLCs about their discussions with, and observations of the mothers: “Content analysis of maternal concerns and breastfeeding problems as documented in the statements of the IBCLCs identified the following themes…”. Throughout the Results of the qualitative data, we clarified the source of the findings by writing “reported/described by the IBCLCs as..”. These statements are highlighted in yellow for easy tracking.

Similarly in the Discussion we clarified that the descriptions of the concerns/participants were based on the IBCLCs statements such as: “This is further supported by the qualitative data reported by the IBCLCs in which they described participants..”

Also in the limitations we added: “Another limitation is the fact that the qualitative data is generated from statements made by the IBCLCs about their discussions with the mothers or observations made during their home visits. Hence the potential for conscious or unconscious subjective bias cannot be ruled out”.

4. Is it possible that there may be (conscious or unconscious) bias in the IBCLCs accounts? Especially as you suggest that the IBCLCs described some women as ‘uncooperative’. The interpretation of the findings therefore needs to be more cautious. 

Answer: We agree with the kind reviewer. The possibility of subjective IBCLC bias cannot be ignored. We included this point as one of the limitations of the study (Please refer to #4 above). 

5. Abstract: The findings are not very informative as they don’t state what frequency, duration etc.. are associated with the outcome. 

Answer: We added the following sentences to the Abstract-Results: “Professional lactation support was received by more mothers continuing six months of EBF (100% vs. 83.3%, p= 0.001). In crude analysis, those mothers had a higher number of overall contacts with lactation experts (mean ± SD of 9.5 ± 2.9 vs. 7.0 ± 4.4, p= 0.001), and longer duration of face-to-face contacts (mean ± SD of 261.9 ± 209.1 vs. 201.0 ± 117.4 minutes, p= 0.035)”. In order to comply with the journal’s word limit requirements, we did not include further details about the number and duration of face-to-face and telephone contacts.

6. Introduction, Line 77: Please use the correct name for the International Code i.e. International Code of Marketing of Breastmilk Substitutes and it should be capitalized.

Answer: Done. Thank you for the alert.

7. Introduction: Lines 90-96 are confusing – what is the difference between participants in the multi-component arm and participants in all 3 intervention components? And what does ‘most engaged with’ mean? Please clarify in the paper.

Answer: In the RCT, not all participants in the experimental group ended up receiving the 3 intervention components (BF education, peer support, lactation support). As in all trials, some participants may not comply with the intervention as dictated by the trial’s protocol. Hence, in our trial, some participants ended up receiving one component, others received 2 components, and a third subgroup received all 3 components. We revised this paragraph as follows: “Participants in the experimental group who complied with all three intervention components were six times more likely than controls to continue EBF for six months (OR=6.63; 95% CI: 3.03 to 14.51). Of the three intervention components, participants were most compliant with professional lactation support (92.6%), received either as a single component, or in combination with prenatal education and/or peer support”. The word ‘engaged’ is replaced by ‘compliant’. 

8. For clarity please refer to breastfeeding mothers (not nursing mothers) throughout the paper.

Answer: Done.

9. Materials & Methods, Line 122: Please change artificial milk to either BMS or infant formula (and throughout the paper).

Answer: Done. 

10. Materials & Methods, Line 123: This is an ambiguous sentence – do you mean there are few IBCLCs who work in private practices and that most work in hospitals? Please clarify.

Answer: We revised this sentence as follows: “There are currently few IBCLCs in Lebanon. They work in private practices and not in hospitals”.

11. Materials & Methods, Lines 150-151: I am not sure these can be described as cultural misconceptions – many women all over the world experience breastfeeding as painful and tiring; and as shameful because of the reaction of others. Please reword to reflect this.

Answer: We removed the word “cultural”. 

12. Materials & Methods, Lines 220-221: The categories not mutually exclusive- in which category does 3 months belong?

Answer: We corrected the intervals to “first postpartum week, second week to third month postpartum and fourth to sixth month postpartum”. Thank you for alerting us to this error. 

13. Results, Table 1: It is unclear what the gestational age row refers to? As it is 17 weeks it can’t be at birth – is it at recruitment? Please clarify. 

Answer: Yes. It was at recruitment. This is corrected in the revised Table 1.

14. Results: You use terms face-to-face and home visits – are these the same? Please explain your use of the terms.

Answer: ‘Face-to-face’ encounter refers to hospital AND home visits, whereas ‘home visits’ refers to home visits only. The following definition was added to the Methods: “Face-to-face support is defined as support delivered during visits to the hospital or to the homes of the participants”. 

15. Discussion, Lines 344-346: This seems to be speculation – is there other evidence you can provide to support your claims?

Answer: We agree with the kind reviewer. We removed the sentence “Consequently, they rejected further IBCLC visits, and did not initiate telephone calls with the IBCLC when needed”.

16. Discussion, Lines 365-367: It could also be that women feel more comfortable to discuss breastfeeding with female obstetricians. 

Answer: In full agreement. We added the following sentence: “Also breastfeeding mothers may feel more comfortable to discuss breastfeeding with female obstetricians”.

We hope we have addressed all the comments of the kind reviewers in a clear manner, and revised the manuscript accordingly.

Thank you again for the thorough review of our paper.

Yours sincerely, 

Mona Nabulsi, MD, MSc

Professor of Clinical Pediatrics

Department of Pediatrics and Adolescent Medicine

Faculty of Medicine

American University of Beirut

Beirut-Lebanon

P.O.Box: 113-6044/C8

Fax: 961-1-370781

 961-1-744464

E-mail: mn04@aub.edu.lb

---

## [Editor Report · Decision Letter 1]

24 Aug 2020

Assessing the impact of professional lactation support frequency, duration and delivery form on exclusive breastfeeding in Lebanese mothers

PONE-D-20-14153R1

Dear Dr. Nabulsi,

We’re pleased to inform you that your manuscript has been judged scientifically suitable for publication and will be formally accepted for publication once it meets all outstanding technical requirements.

Kind regards,

Jane Anne Scott, PhD, MPH Grad Dip Dietetics, BSc

Academic Editor

PLOS ONE
---

## [Editor Report · Acceptance letter]

28 Aug 2020

PONE-D-20-14153R1 

Assessing the impact of professional lactation support frequency, duration and delivery form on exclusive breastfeeding in Lebanese mothers 

Dear Dr. Nabulsi:

I'm pleased to inform you that your manuscript has been deemed suitable for publication in PLOS ONE. Congratulations! Your manuscript is now with our production department. 

Kind regards, 

on behalf of

Dr. Jane Anne Scott 

Academic Editor

PLOS ONE